# A Comparison of an Australian First Nations Primary Healthcare Data Specification with Potentially Preventable Hospitalisations

**DOI:** 10.3390/ijerph21091192

**Published:** 2024-09-09

**Authors:** Boyd Potts, Christopher M. Doran, Stephen J. Begg

**Affiliations:** 1Cluster for Resilience and Wellbeing, Appleton and Manna Institutes, Central Queensland University, Brisbane, QLD 4701, Australia; b.potts@cqu.edu.au; 2Violet Vines Marshman Centre for Rural Health Research, La Trobe University, Bendigo, VIC 3550, Australia; s.begg@latrobe.edu.au

**Keywords:** First Nations peoples, potentially preventable hospitalisations, data specification

## Abstract

Potentially Preventable Hospitalisations (PPH) is a widely used indicator of the effectiveness of non-hospital care. Specified using the International Classification of Diseases (ICD) coding, PPH comprises a suite of health conditions that could have potentially been prevented with appropriate care. The most recent edition of the *National Guide to a Preventative Health Assessment for First Nations People* documents the health conditions of interest to providers of primary care, many of which are not represented in PPH. Given the *National Guide* has been developed specifically with First Nations in mind, the aim of this research is twofold. The first aim is to formally posit the question of whether a summative measure of hospitalisations aligned diagnostically to the *National Guide* has value either as an alternative or complement to PPH in the context of First Nations primary health information. The second aim is to develop and present a prototype *ICD-10* data specification for such a measure, referred to as the First Nations primary healthcare (FNPHC) data specification, and examine the age-standardised hospitalisation rates for FNPHC and PPH for correlations and/or differences. Age-standardised hospitalisation rates from 2016–17 to 2019–20 using both classifications were examined to assess the usefulness and relevance of summative measures of hospitalisations for informing primary care. Rates of FNPHC for principal diagnoses were between 1.5 and 2.5 times higher than those of PPH and approximately between 6 and 12 times higher for additional diagnoses. There was a strong correlation with PPH when rates were compared across all observations: jurisdictions with higher rates of PPH tended to have higher rates of hospitalisations according to the custom specification. Findings support its application as a summary measure for First Nations primary care providers. Given the policy landscape in Australia that aims to close the gap, it is imperative that measures of primary health take advantage of the concepts and application of First Nations data sovereignty and governance. The validity and cultural appropriateness of the First Nations primary health data specification needs to be further researched.

## 1. Introduction

In 2007, the Australian Commonwealth, state, territory and local governments made a commitment to work together to close the gap in First Nations disadvantage [1]. This led to the National First Nations Reform Agreement, a significant step toward more co-ordinated action [2]. There are 17 national socio-economic targets across areas that have impacts on life outcomes for First Nations Australians in health and wellbeing, education, employment, justice, safety, housing, land and waters, languages and digital inclusion. Five of these targets relate to health and well-being, with a key priority being to close the health and life-expectancy gap between First Nations and other Australians within a generation [1]. Achieving equality in life expectancy and closing the gap in life expectancy within a generation is not on track to be met by 2031: First Nations people still have a lower life expectancy than other Australians [2,3].

The primary care sector plays an integral and determining role in achieving these national targets [4,5,6,7,8,9,10]; however, the data and information available on the health and well-being of Australian people are predominantly sourced from outside the primary care sector. Since 2006, the Australian Institute of Health and Welfare (AIHW) has generated Health Performance Framework (HPF) reports that provide a range of information pertaining to First Nations health and well-being outcomes, as well as factors influencing the performance of the health system [7,8,9]. Data from hospital admissions and national health surveys feature heavily in this framework. A recent article published in this journal questioned the utility of the HPF to inform healthcare reform [11]. Hospitalisation for ambulatory care-sensitive conditions, also called ‘potentially preventable’ or ‘avoidable’ hospitalisations, has been used extensively as an indicator of the accessibility and overall effectiveness of primary healthcare [12,13,14]. Specified using the *International Classification of Diseases, 10th Edition, Australian Modification* (*ICD-10-AM*), Potentially Preventable Hospitalisations (PPH) are organised into 22 diagnosis groups across 3 categories of vaccine-preventable conditions, chronic conditions and acute conditions [15].

Latest data from Australia suggest that, in 2022–2023, there were 548,000 PPHs in public hospitals and 178,000 in private hospitals. These included 273,000 hospitalisations for chronic conditions (excluding diabetes), 58,300 hospitalisations for diabetes complications, and 55,500 hospitalisations for vaccine-preventable conditions. The likelihood of a person having a PPH can vary according to their age, sex, where they live, their level of socioeconomic disadvantage and First Nations status [8]. First Nations people account for 3.3% of the Australian population [16] but were hospitalised for potentially preventable conditions at three times the rate of non-First Nation Australians between July 2019 and June 2021 (based on age-standardised rates) [9]. Overall, between 2013–2014 and 2020–2021, the gap between First Nations people and First Nation Australians increased from a rate difference of 40 PPH per 1000 population (in 2013–2014) to 46 PPH per 1000 population (in 2020–2021). At a national level, the rate of PPH for First Nations people was higher for those living in remote rather than non-remote areas. The rate among First Nations people was highest for those living in remote areas (97 PPH per 1000 population), followed by very remote areas (91 PPH per 1000 population). The rate was lowest for those in major cities and inner regional areas (both 40 PPH per 1000 population), followed by outer regional areas (57 per 1000 population) [9].

The majority of PPHs involve conditions that could have been identified and treated earlier by either primary healthcare or public health interventions and, thus, prevented, or at least limited, the necessity for hospital care. Consequently, the indicator is commonly used by governments to provide an evidence-based foundation for targeted interventions designed to control costs and improve primary healthcare effectiveness. A review of the Health Needs Assessments published by Australia’s 31 primary health networks found that PPH was included as an indicator of primary health needs in all assessments that were publicly accessible (n = 29). Approximately two-thirds (n = 19) referenced PPH specifically in the context of First Nations primary health needs [3,17,18,19,20,21,22,23,24,25,26,27,28,29,30,31,32].

The most recent edition of the *National Guide* to a Preventative Health Assessment for First Nations people documents the health conditions of interest to providers of primary care [33]. It compiles comprehensive, evidence-based advice and guidance on the best practices for providers of primary healthcare across the lifecycle. Developed with the National Aboriginal Community Controlled Health Organisation and the Royal Australian College of General Practitioners, the *National Guide* consists of seventeen chapters, each addressing an important domain of health and wellbeing. Despite the dominant use of PPH in health statistics as a proxy measure of primary care effectiveness, PPH omits many of the health conditions relevant to primary care according to the *National Guide* (e.g., cancer, chronic kidney disease, conditions of the eyes and ears, STI, mental health, alcohol and drug use, dementia, osteoporosis, lifestyle factors) [33]. Given the *National Guide* has been developed specifically with First Nations primary health in mind, the aim of this research is twofold. The first aim is to formally posit the question of whether a summative measure of hospitalisations aligned diagnostically to the *National Guide* has value either as an alternative or complement to PPH in the context of First Nations primary health information. The second aim is to develop and present a prototype *ICD-10* data specification for such a measure, referred to as the First Nations primary healthcare (FNPHC) data specification, and examine the age-standardised hospitalisation rates for FNPHC and PPH for correlations and/or differences.

## 2. Methods

### 2.1. Approach

The *National Guide* was reviewed, and health conditions of interest were coded using the *ICD-10-AM* (referred to interchangeably hereafter with *ICD-10* for brevity where needed). Where a condition also existed as a measure in the HPF reporting, the *ICD-10* specifications utilised and published by the Australian Institute of Health and Welfare were adopted. For example, Chapter 9 of the *National Guide* relates to respiratory health with a subsection specifically for asthma. Section 1.04 of the HPF relates to respiratory health, and the *ICD-10* specification for asthma utilised in HPF reporting (J45–J46) was adopted for the FNPHC specification also. Conversely, Chapter 5 of the *National Guide* relates to the health of older persons and has a subsection for falls in people 50 years and older. No such specification exists in the HPF, so the relevant *ICD-10* codes W05–W10 were included in this category. Figure 1 summarises this decision process.

*ICD-10* codes were organised into sixteen categories: Diabetes, Respiratory health, Circulatory health, Acute rheumatic fever and rheumatic heart disease (ARF/RHD), Chronic kidney disease (CKD), Eyes and ears, Oral health, Sexually transmitted infections (STI) and blood-borne diseases, Cancer, Depression and suicide, Mental disorders, Mental-health related conditions, Alzheimer and dementia, Alcohol and drugs, Older adults and Lifestyle factors. Age-standardised hospitalisation rates from 2016–2017 to 2019–2020 using both classifications were calculated. FNPHC rates were compared descriptively with PPH with respect to correlations and rate ratios. Hospitalisation rates using the FNPHC specification were examined for each diagnosis category, principal or additional diagnoses and regional variation to explore an application of the FNPHC and consider implications for implementation in practice.

### 2.2. Data

Data were requested from the National Hospital Morbidity Database—a repository of episode-level records from admitted patient data collections in Australian public and private hospitals—using the FNPHC and PPH specifications. Annual counts of admitted patients (identified as First Nations people) were provided for each specified category for financial years 2016–2017 through 2019–2020 by State/Territory and five–year age groups from 0–4 through 65+. Counts were provided for both principal (PDx) and additional (ADx) diagnoses under the FNPHC specification, except for codes designating external causes (X60–X84, X45, Y15, W05–W10, W18–W19, X85–Y09) for which the concept of principal or additional diagnosis does not apply.

Population data for First Nations people were obtained from the Australian Bureau of Statistics via the online data explorer (explore.data.abs.gov.au, accessed on 22 December 2022), using the Series B projection for States and Territories. Population data by calendar year was converted to financial years by taking the average of the population across the two calendar years.

Age-standardised hospitalisation rates (per 1000 population) were calculated using the 2001 Australian population standard for both FNPHC and PPH specifications. For the FNPHC specification, rates for both principal and additional diagnoses for all categories and the sum total were calculated and compared. External cause codes were included in the relevant category totals (Table 1) for both (principal/additional) calculations. External cause diagnoses were included in the relevant category totals for both principal and additional diagnosis rate calculations.

### 2.3. Ethics

Ethics approval was obtained from the Human Research Ethics Committee at Central Queensland University (Application Number 22739).

## 3. Results

Table 1 shows the FNPHC specification and its alignment with the *National Guide* and HPF. There was good alignment between these categories and the chapters of the *National Guide*, particularly where the relevant health conditions are presented as discrete chapters, for example, Respiratory health (Chapter 9). Conversely, some conditions were relevant to more than one chapter. Diabetes, for example, is important to antenatal care (Chapter 2) and diabetes prevention and early detection (Chapter 12). The category of Lifestyle factors contained several items that were important to more than one chapter of the *National Guide*.

Extensions to the chapters were made in developing the FNPHC specification. Chapter 17 of the *National Guide* addresses mental health but specifies only prevention of depression and prevention of suicide as sub–chapters. Together, these comprise the Depression and suicide category in the FNPHC specification to align with the *National Guide*. The categories Mental disorders and Mental health-related conditions were adopted directly from the HPF and included in the FNPHC specification (excluding those codes already used for depression and dementia in their respective categories) to provide greater utility in measuring mental health more broadly. Chapter 16 relates to family abuse and violence, for which no *ICD-10* specifications could be established. External cause codes for assault (X85–Y09) were included in the FNPHC specification under Lifestyle factors to capture this aspect. Implications are considered in the discussion.

### 3.1. FNPHC Categories

As a total measure, FNPHC hospitalisations with a principal diagnosis at the national level were between 125.5 and 134.7 per 1000 for the period of collected data. Hospitalisations for FNPHC as an additional diagnosis are over four times higher each year—between 561.1 and 597.3 per 1000. Additional diagnoses were more inclusive in all categories except for Respiratory health, Eyes and ears and Oral health, where rates of principal diagnoses were between 1.3 and 1.7 times higher than those of additional diagnoses. National rates by year are presented for all FNPHC categories in Table 2.

### 3.2. Comparison with Potentially Preventable Hospitalisations

Age-standardised FNPHC rates were compared with those of PPH. FNPHC and PPH specifications are not independent. *ICD-10* codes common to both specifications are shown in Appendix A. Little variability of rates across years was observed in all jurisdictions for both PPH and FNPHC measures (see Figure 2). Rates of FNPHC as principal diagnoses were between 1.5 and 2.5 times higher than those of PPH and approximately between 6 and 12 times higher for additional diagnoses (Figure 2 and Figure 3). Rates for additional diagnoses increased differentially for Western Australia compared to other jurisdictions owing to an increase in additional diagnoses in the categories Diabetes and CKD, peaking in 2018–2019. Rates and ratios for all years by jurisdiction are included in Appendix A.

Spearman’s rank correlation coefficients between measures were not significant except for South Australia, which had a negative correlation (years with higher rates of PPH had lower rates of FNPHC; ρ = −1, *p* < 0.001), and for the Australian Capital Territory, which had a positive correlation (years with higher rates of PPH also had higher rates of FNPHC; ρ = 1, *p* < 0.001) for principal diagnoses only (Table 3). Correlating all observations by jurisdiction-year showed a positive association with PPH in both principal and additional diagnoses: jurisdictions with higher rates of PPH were statistically likely to also have higher rates of FNPHC (ρ = 0.95–0.96, *p* < 0.001; see Figure 3).

### 3.3. Principal and Additional Diagnoses

As a total measure, FNPHC rates for additional diagnoses were higher than rates for principal diagnoses. This was also true of most individual categories, except for Oral health, Eyes and ears and Respiratory health, which were more likely to be reported as principal diagnoses. STI and blood-borne diseases were almost exclusively reported in additional diagnoses in Tasmania and the ACT and accounted for less than one per thousand hospitalisations in other jurisdictions. Figure 4 shows the rate ratios of additional to principal diagnoses for each category and financial year. All rates by jurisdiction, year and category are provided in Appendix A.

Rates based on principal diagnoses were correlated with those based on additional diagnoses for all FNPHC categories by State/Territory and financial year. There was a generally high-ranking correlation between rates based on principal and additional diagnoses for each category. That is, jurisdictions and financial years with high rates of principal diagnoses also tended to have higher rates of additional diagnoses relative to other jurisdictions and years. Values of Spearman’s correlation were high, between 0.7 and 0.9, except for Mental health-related conditions (ρ = 0.44), and were significant at the 0.01 level. However, weak or non-existent correlations were observed for Oral health, Cancer, Depression and suicide, Older adults and Lifestyle factors, and none of which were statistically significant. Table 4 shows the rank correlation coefficients for each category.

## 4. Discussion

This study developed a prototype *ICD-10* specification to reflect the health conditions of interest to First Nations primary healthcare as advised by the *National Guide* and compared rates of hospitalisation with those of the traditional PPH data. As a total summative measure, the FNPHC classification contains more diagnostic categories than PPH, and this is reflected in the higher rates of hospitalisation observed under the former classification.

There was no clear evidence of a correlation between the two measures; however, it must be noted that hospitalisation rates nationally changed by less than 5 per 1000 for PPH and less than 10 per 1000 for FNPHC across the period observed. There was a strong correlation between the measures for jurisdiction-years, with States/Territories that had higher rates of PPH also tended to have higher rates of FNPHC hospitalisations. The specifications are not mutually exclusive in terms of the diagnoses included, which contributes to the statistical association; however, many of the conditions documented in the *National Guide* are not represented in the PPH specification. It is, therefore, also possible that both specifications reflect the underlying area’s propensity for hospitalisations or health status in a general way. Analysis of data for smaller geographical regions and/or linked data for individuals would help elucidate these relationships.

Year-to-year variability was more likely to be observed when considering additional diagnoses as the inclusion criteria rather than principal diagnoses alone. With respect to individual disease/condition categories, higher rates were typically observed in additional diagnoses; however, this was not the case for Oral health, Eyes and ears and Respiratory health. Further consultation is required with First Nations health workers and health practitioners regarding the appropriateness and usefulness of principal or additional diagnoses. Further, as this work is a desktop and document analysis, clinical validation and further refinements to the specification would strengthen the validity of the measure, such as those benefiting the PPH specification: diagnosis-specific inclusion/exclusion criteria regarding principal or additional diagnoses, age and hospital procedures. For example, heart failure (I50) is counted only as a principal diagnosis and excludes admissions for heart surgeries; acute bronchitis (J20) is counted only as a principal diagnosis and if the admission has an additional diagnosis of bronchiectasis (J47). Similar refinements to the FNPHC specification warrant consideration.

Of particular importance is the fact that cancer is not included in the PPH specification but has a dedicated chapter in the *National Guide*. This highlights the fundamental disconnect between the concept of avoidable hospitalisations and the prevalence of health conditions that are of interest to primary care that motivated the current study. The FNPHC specification has high face validity owing to its direct relationship to the *National Guide*; however, the value of a broad, summative measure of hospitalisations to inform First Nations primary care—be it PPH or FNPHC—should also be questioned and further tested. As with any indicator, they can add value when used as outcome variables in conjunction with other explanatory or input variables, such as those reflecting service activity in the case of primary care, rather than providing a numerical value in isolation. Examining correlations between FNPHC and other health system measures is a topic for future research.

## 5. Conclusions

This research has developed a First Nations primary health data (FNPHC) specification and compared it with the traditional specification derived using PPH. Preliminary findings support its application as a summary measure for First Nations primary care providers to monitor hospitalisations for health conditions specified in the *National Guide*. Given the policy landscape in Australia that aims to close the gap, it is imperative that measures of primary health take advantage of the concepts and application of First Nations data sovereignty and governance. The validity and cultural appropriateness of the First Nations primary health data specification needs to be further researched.

## Figures and Tables

**Figure 1 ijerph-21-01192-f001:**
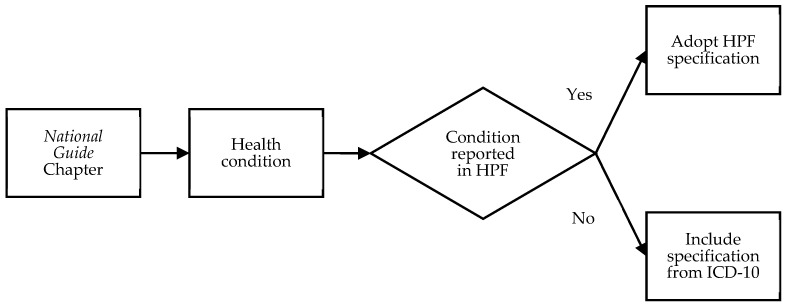
Process for *ICD-10-AM* specification of conditions addressed in the *National Guide*.

**Figure 2 ijerph-21-01192-f002:**
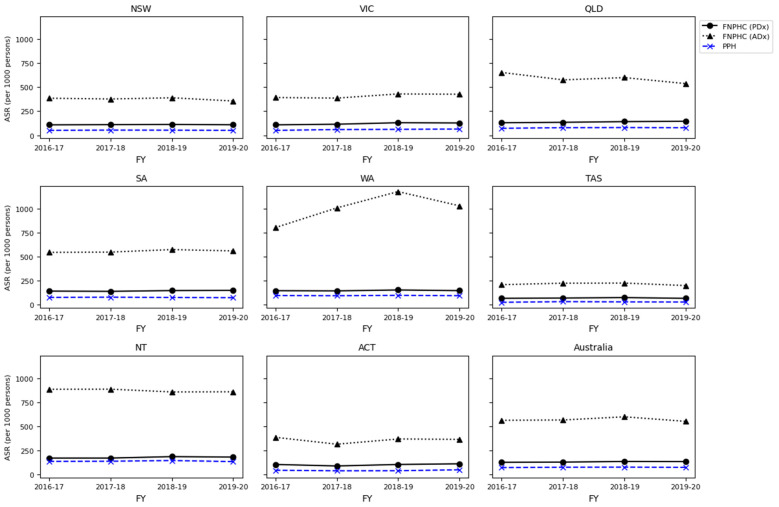
Jurisdictional comparison of PPH and FNPHC hospitalisation rates.

**Figure 3 ijerph-21-01192-f003:**
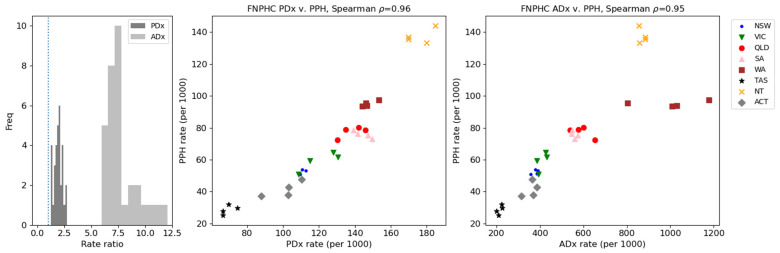
Rate ratios and correlation between all observations of PPH and FNPHC hospitalisations. The distribution of rate ratios across State/Territory for FNPHC:PPH is shown; a vertical line denotes ratio = 1.

**Figure 4 ijerph-21-01192-f004:**
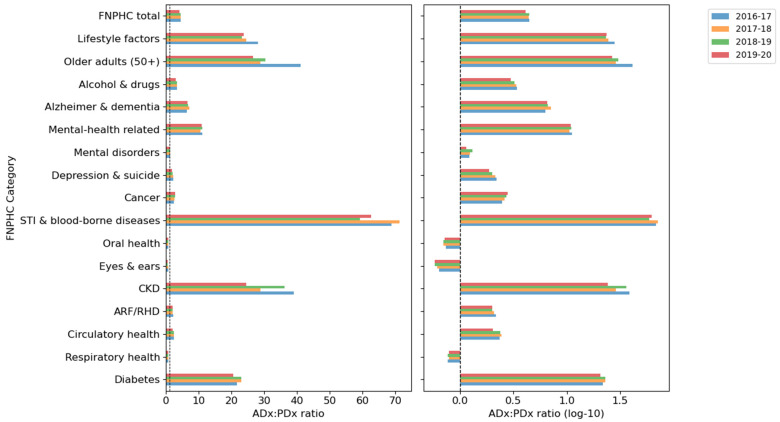
Rate ratios of additional diagnoses and principal diagnoses for FNPHC categories. The second panel shows ratios on a base-10 logarithmic scale to highlight categories where principal diagnoses exceed additional diagnoses.

**Table 1 ijerph-21-01192-t001:** *ICD-10-AM* specification for conditions of interest to First Nations primary healthcare (FNPHC).

Category	Sub-Category	ICD-10 Code(s)	ICD-10 Description	National Guide Alignment	Health Performance Framework Alignment
Diabetes	Diabetes	E10–E14	Diabetes mellitus	Chapter 2: Antenatal care-Diabetes, Chapter 12: Type 2 diabetes prevention and early detection-Diabetes	1.09 Diabetes, 1.23 Leading causes of mortality, 3.05 Chronic disease management, 3.18 Care planning for chronic diseases
Respiratory health	Asthma	J45	Asthma	Chapter 9: Respiratory health-Asthma	1.04 Respiratory health
		J46	Status asthmaticus		
	COPD	J41–J44	Simple and mucopurulent chronic bronchitis; Unspecified chronic bronchitis; Emphysema; Other chronic obstructive pulmonary disease	Chapter 9: Respiratory health-Chronic obstructive pulmonary disease	1.04 Respiratory health, 3.02 Immunisation
	Bronchiectasis	J47	Bronchiectasis	Chapter 9: Respiratory health-Bronchiectasis and chronic suppurative lung disease	1.04 Respiratory health, 3.02 Immunisation
	Influenza	J09–J11	Influenza	Chapter 9: Respiratory health-Influenza prevention	1.04 Respiratory health, 3.02 Immunisation
	Pneumonia	J12–J18	Pneumonia	Chapter 9: Respiratory health-Pneumococcal disease prevention	1.04 Respiratory health, 3.02 Immunisation
Circulatory health	Other circulatory disease	I60–I69	Cerebrovascular diseases	Chapter 11: Cardiovascular disease prevention-CVD	1.23 Leading causes of mortality, 1.05 Circulatory disease
		I70–I79	Diseases of arteries, arterioles and capillaries		
		I80–I89	Diseases of veins, lymphatic vessels and lymph nodes not elsewhere classified		
		I95–I99	Other and unspecified disorders of the circulatory system		
	Heart disease	I20–I25	Ischaemic heart diseases	Chapter 11: Cardiovascular disease prevention-CVD	1.23 Leading causes of mortality, 1.05 Circulatory disease
		I26–I28	Pulmonary heart disease and diseases of pulmonary circulation		
		I30–I52	Other forms of heart disease		
	Hypertension	I10–I15	Hypertensive diseases	Chapter 11: Cardiovascular disease prevention-CVD	1.23 Leading causes of mortality, 1.05 Circulatory disease, 1.07 High blood pressure,
ARF/RHD	ARF	I00–I02	Acute rheumatic fever	Chapter 10: Acute rheumatic fever and rheumatic heart disease	1.06 Acute rheumatic fever and rheumatic heart disease
	RHD	I05–I09	Chronic rheumatic heart diseases	Chapter 10: Acute rheumatic fever and rheumatic heart disease	1.06 Acute rheumatic fever and rheumatic heart disease
CKD	CKD	N18	Chronic kidney disease	Chapter 13: Chronic kidney disease prevention and management-Kidney disease	1.10 Kidney disease, 1.23 Leading causes of mortality
Eyes and Ears	Trachoma and trichiasis	A71	Trachoma	Chapter 6: Eye health-Trachoma and trichiasis	1.16 Eye health
		H02.0	Entropion and trichiasis of eyelid		
	Eyes	H00–H59	Diseases of the eye and adnexa	Chapter 6: Eye health-Visual acuity	1.16 Eye health
	Ears	H60–H95	Diseases of the ear and mastoid process	Chapter 7: Hearing loss-Hearing	1.15 Ear health
Oral health	Oral health	K02	Dental caries	Chapter 8: Oral and dental health-Oral and dental	1.11 Oral health
		K05	Gingivitis and periodontal diseases		
		K08.1	Complete loss of teeth		
STI and blood-borne diseases	Hepatitis C	B17.1	Acute hepatitis C	Chapter 14: Sexual health and blood-borne viruses-Blood-borne viruses, Chapter 2: Antenatal care-Genitourinary and blood-borne viral infections	3.02 Immunisation
		B18.2	Chronic viral hepatitis C		
	Hepatitis	B17.1	Acute hepatitis C	Chapter 14: Sexual health and blood-borne viruses-Blood-borne viruses, Chapter 2: Antenatal care-Genitourinary and blood-borne viral infections	3.02 Immunisation
		B18.2	Chronic viral hepatitis C		
		B16	Acute hepatitis B	Chapter 14: Sexual health and blood-borne viruses-Blood-borne viruses	3.02 Immunisation
		B17.0	Acute delta-(super)infection of hepatitis B carrier		
		B18.0	Chronic viral hepatitis B with delta-agent		
		B18.1	Chronic viral hepatitis B without delta-agent		
	Sexual health	A50–A64	Infections with a predominantly sexual mode of transmission	Chapter 14: Sexual health and blood-borne viruses-Sexually transmitted infections, Chapter 2: Antenatal care-Genitourinary and blood-borne viral infections	1.12 HIV, AIDS, hepatitis and sexually transmissible infections
		B20–B24	Human immunodeficiency virus [HIV] disease		
Cancer	Cancer	C50	Malignant neoplasm of breast	Chapter 15: Prevention and early detection of cancer-Prevention and early detection of breast cancer	1.08 Cancer, 1.23 Leading causes of mortality
		C53	Malignant neoplasm of cervix uteri	Chapter 15: Prevention and early detection of cancer-Prevention and early detection of cervical cancer	
		C18	Malignant neoplasm of colon	Chapter 15: Prevention and early detection of cancer-Prevention and early detection of colorectal (bowel) cancer	
		C19	Malignant neoplasm of rectosigmoid junction		
		C20	Malignant neoplasm of rectum		
		C21.8	Malignant neoplasm: Overlapping lesion of rectum, anus and anal canal		
		C22	Malignant neoplasm of liver and intrahepatic bile ducts	Chapter 15: Prevention and early detection of cancer-Prevention and early detection of primary liver (hepatocellular) cancer	
		C34	Malignant neoplasm of bronchus and lung	Chapter 15: Prevention and early detection of cancer-Prevention of lung cancer	
		C61	Malignant neoplasm of prostate	Chapter 15: Prevention and early detection of cancer-Early detection of prostate cancer	
Depression and suicide	Depression	F32	Depressive episode	Chapter 17: Mental health-Prevention of depression	3.10 Access to mental health services
		F33	Recurrent depressive disorder		
		F34.1	Dysthymia		
	Suicide and Intentional Self-harm	X60–X84	Intentional self-harm	Chapter 17: Mental health-Prevention of suicide	3.10 Access to mental health services, 1.23 Leading causes of mortality
Mental disorders	Mental disorders (not due to substance abuse)	F04–F09	Organic, including symptomatic mental disorders (excluding dementia codes F00–F03).	Chapter 17: Mental health-Mental disorders (not specified in guidelines)	3.10 Access to mental health services
		F20–F29	Schizophrenia, schizotypal and delusional disorders		
		F30–F39	Mood [affective] disorders (excluding depression codes F32, F33, F34.1).		
		F40–F48	Neurotic, stress-related and somatoform disorders		
		F50–F59	Behavioural syndromes associated with physiological disturbances and physical factors		
		F60–F69	Disorders of adult personality and behaviour		
		F70–F79	Mental retardation		
		F80–F89	Disorders of psychological development		
		F90–F98	Behavioural and emotional disorders with onset usually occurring in childhood and adolescence		
		F99	Unspecified mental disorder		
Mental-health-related conditions	Mental health-related conditions	G47.0	Disorders of initiating and maintaining sleep [insomnias]	Chapter 17: Mental health-Mental health-related hospitalisations (not specified in guidelines), Chapter 4: The health of young people-Social emotional wellbeing	1.18 Social and emotional well-being, 3.10 Access to mental health services
		G47.1	Disorders of excessive somnolence [hypersomnias]		
		G47.2	Disorders of the sleep–wake schedule		
		G47.8	Other sleep disorders		
		G47.9	Sleep disorder, unspecified		
		O99.3	Mental disorders and diseases of the nervous system complicating pregnancy, childbirth and the puerperium		
		R44	Other symptoms and signs involving general sensations and perceptions		
		R45.0	Nervousness		
		R45.1	Restlessness and agitation		
		R45.4	Irritability and anger		
		R48	Dyslexia and other symbolic dysfunctions, nec		
		Z00.4	General psychiatric examination, not elsewhere classified		
		Z03.2	Observation for suspected mental and behavioural disorders		
		Z04.6	General psychiatric examination, requested by authority		
		Z09.3	Follow-up examination after psychotherapy		
		Z13.3	Special screening examination for mental and behavioural disorders		
		Z50.2	Alcohol rehabilitation		
		Z50.3	Drug rehabilitation		
		Z54.3	Convalescence following psychotherapy		
		Z61.9	Negative life event in childhood, unspecified		
		Z63.1	Problems in relationship with parents and in-laws		
		Z63.8	Other specified problems related to primary support group		
		Z63.9	Problem related to primary support group, unspecified		
		Z65.8	Other specified problems related to psychosocial circumstances		
		Z65.9	Problem related to unspecified psychosocial circumstances		
		Z71.4	Alcohol abuse counselling and surveillance		
		Z76.0	Issue of repeat prescription		
Alzheimer and dementia	Alzheimer	G30	Alzheimer’s disease	Chapter 17: Mental health–Alzheimer (not specified in guidelines)	3.10 Access to mental health services, 1.23 Leading causes of mortality
	Dementia	F00–F03	Dementia in Alzheimer’s disease; Vascular dementia; Dementia in other diseases classified elsewhere; Unspecified dementia	Chapter 5: The health of older people-Dementia	3.10 Access to mental health services
Alcohol and drugs	Alcohol	F10	Mental and behavioural disorders due to use of alcohol	Chapter 1: Lifestyle-Alcohol	2.16 Risky alcohol consumption, 3.11 Access to alcohol and drug services
		T51	Toxic effect of alcohol		
		X45	Accidental poisoning by and exposure to alcohol		
		Y15	Poisoning by and exposure to alcohol, undetermined intent		
	Drugs	F11–F16	Mental and behavioural disorders due to use of opioids; Mental and behavioural disorders due to use of cannabinoids; Mental and behavioural disorders due to use of sedatives or hypnotics; Mental and behavioural disorders due to use of cocaine; Mental and behavioural disorders due to use of other stimulants, including caffeine; Mental and behavioural disorders due to use of hallucinogens	Chapter 4: The health of young people-Illicit drug use	2.17 Drug and other substance use, including inhalants; 3.11 Access to alcohol and drug services
		F18	Mental and behavioural disorders due to use of volatile solvents		
		F19	Mental and behavioural disorders due to multiple drug use and use of other psychoactive substances		
		T36	Poisoning by systemic antibiotics		
		T37	Poisoning by other systemic anti-infectives and antiparasitics		
		T39	Poisoning by nonopioid analgesics, antipyretics and antirheumatics		
		T40	Poisoning by narcotics and psychodysleptics [hallucinogens]		
		T42	Poisoning by antiepileptic, sedative–hypnotic and anti-Parkinsonism drugs		
		T43	Poisoning by psychotropic drugs, not elsewhere classified		
		T52	Toxic effect of organic solvents		
Older adults (50+)	Falls	W05–W10	Fall involving wheelchair; Fall involving bed; Fall involving chair; Fall involving other furniture; Fall involving playground equipment; Fall on and from stairs and steps	Chapter 5: The health of older people-Falls	
		W18–W19	Other fall on same level; Unspecified fall		
	Osteoporosis	M80–M82	Osteoporosis with pathological fracture; Osteoporosis without pathological fracture; Osteoporosis in diseases classified elsewhere	Chapter 5: The health of older people-Osteoporosis	
Lifestyle factors	Anaemia	D50–D53	Nutritional anaemias	Chapter 3: Child health-Anaemia, Chapter 2: Antenatal care-Nutrition and nutritional supplementation	2.19 Dietary behaviour
	Obesity	E66	Obesity	Chapter 1: Lifestyle-Overweight and obesity	2.22 Overweight and obesity
	Assault	X85–Y09	Assault	Chapter 16: Family abuse and violence-Violence (Assault not specified in guidelines except for FAV)	2.10 Community safety
	Tobacco use	Z72.0	Tobacco use	Chapter 1: Lifestyle-Smoking, Chapter 2: Antenatal care-Smoking cessation	2.21 Healthy behaviours during pregnancy, 2.03 Environmental tobacco smoke, 2.15 Tobacco use
	Alcohol use	Z72.1	Alcohol use	Chapter 1: Lifestyle-Alcohol	2.16 Risky alcohol consumption, 3.11 Access to alcohol and drug services
	Drug use	Z72.2	Drug use	Chapter 4: The health of young people-Illicit drug use	2.17 Drug and other substance use, including inhalants; 3.11 Access to alcohol and drug services
	Physical activity	Z72.3	Lack of physical exercise	Chapter 1: Lifestyle-Physical activity	2.18 Physical activity
	Diet	Z72.4	Inappropriate diet and eating habits	Chapter 2: Antenatal care-Nutrition and nutritional supplementation, Chapter 1: Lifestyle-Dietary habits (not chapter-specific)	2.19 Dietary behaviour, 1.23 Leading causes of mortality, 2.19 Dietary behaviour
	Sexual behaviour	Z72.5	High-risk sexual behaviour	Chapter 14: Sexual health and blood-borne viruses-General prevention advice	1.12 HIV, AIDS, hepatitis and sexually transmissible infections
	Gambling	Z72.6	Gambling and betting	Chapter 1: Lifestyle-Gambling	
	Other lifestyle	Z72.8	Other problems related to lifestyle	Chapter 1: Lifestyle-Other lifestyle problems (not chapter-specific)	
	Lifestyle, unspecified	Z72.9	Problem related to lifestyle, unspecified	Chapter 1: Lifestyle-Unspecified lifestyle problems (not chapter-specific)	

**Table 2 ijerph-21-01192-t002:** Age-standardised hospitalisation rates for FNPHC for financial years 2016–2017 through 2019–2020.

	FY 2016–2017			FY 2017–2018			FY 2018–2019			FY 2019–2020		
FNPHC Category	PDx	ADx	Any Dx	PDx	ADx	Any Dx	PDx	ADx	Any Dx	PDx	ADx	Any Dx
Diabetes	6.15	133.66	139.81	6.48	148.69	155.16	7.09	162.72	169.81	7.13	146.91	154.04
Respiratory health	26.03	20.02	46.04	27.31	21.39	48.71	28.74	22.15	50.89	26.38	20.80	47.17
Circulatory health	30.54	71.42	101.96	29.81	73.04	102.85	30.94	73.89	104.84	30.47	61.88	92.35
ARF/RHD	0.82	1.79	2.61	0.81	1.68	2.48	0.84	1.69	2.54	0.72	1.44	2.15
CKD	1.73	67.56	69.29	1.80	51.78	53.58	1.73	62.49	64.21	2.01	49.05	51.06
Eyes and ears	14.08	8.95	23.03	14.32	8.80	23.12	15.42	8.97	24.39	13.95	8.09	22.04
Oral health	1.80	1.33	3.13	1.88	1.32	3.20	2.02	1.41	3.43	1.83	1.32	3.15
STI and blood-borne diseases	0.32	21.74	22.05	0.30	21.57	21.87	0.35	20.51	20.86	0.31	19.25	19.56
Cancer	4.08	10.09	14.17	4.04	10.52	14.56	4.38	11.94	16.32	4.42	12.38	16.80
Depression and suicide	3.31	7.30	10.61	3.26	7.03	10.29	3.59	7.21	10.80	3.63	6.81	10.44
Mental disorders	14.59	17.76	32.36	15.31	18.97	34.28	16.09	21.01	37.10	17.80	20.31	38.12
Mental-health related	0.71	7.92	8.63	0.80	8.40	9.20	0.87	9.65	10.52	0.79	8.61	9.40
Alzheimer and dementia	0.56	3.55	4.12	0.46	3.24	3.70	0.50	3.34	3.84	0.52	3.44	3.96
Alcohol and drugs	16.08	54.78	70.85	15.87	53.40	69.27	16.24	52.83	69.07	17.82	53.16	70.98
Older adults (50+)	0.09	3.85	3.94	0.14	3.97	4.11	0.13	4.01	4.14	0.16	4.13	4.28
Lifestyle factors	4.62	129.38	134.01	5.32	130.17	135.49	5.75	133.48	139.24	5.56	132.24	137.80
FNPHC total	125.52	561.08	686.60	127.90	563.97	691.87	134.68	597.30	731.99	133.49	549.81	683.30

**Table 3 ijerph-21-01192-t003:** Age-standardised rates of Potentially Preventable Hospitalisations and rank correlations with FNPHC hospitalisation rates, States/Territories.

					Spearman’s Rank Correlation			
	Age-Standardised Hospitalisation Rate (per 1000)				PDx		ADx	
State/Territory	2016–2017	2017–2018	2018–2019	2019–2020	rho	*p*	rho	*p*
NSW	51.12	53.90	53.11	51.02	0.60	0.40	0.40	0.60
VIC	51.02	59.36	61.49	64.66	0.80	0.20	0.60	0.40
QLD	72.50	78.83	80.26	78.41	0.40	0.60	−0.20	0.80
SA	76.23	78.47	75.42	73.15	−1.00	<0.001	−0.60	0.40
WA	95.65	93.50	97.55	93.93	0.80	0.20	0.40	0.60
TAS	25.04	32.01	29.66	27.76	0.60	0.40	0.60	0.40
NT	135.53	136.82	144.14	133.11	0.40	0.60	−0.20	0.80
ACT	42.82	37.11	37.76	47.66	1.00	<0.001	0.40	0.60
Australia	70.91	74.61	75.62	73.22	0.80	0.20	0.80	0.20

**Table 4 ijerph-21-01192-t004:** Spearman’s rank correlation between principal and additional diagnoses for all observations.

FNPHC Category	ρ	*p*
Diabetes	0.79	<0.001
Respiratory health	0.96	<0.001
Circulatory health	0.77	<0.001
ARF/RHD	0.91	<0.001
CKD	0.77	<0.001
Eyes and ears	0.84	<0.001
Oral health	−0.04	0.81
STI and blood-borne diseases	0.78	<0.001
Cancer	0.14	0.44
Depression and suicide	0.17	0.36
Mental disorders	0.87	<0.001
Mental-health related	0.44	0.01
Alzheimer’s and dementia	0.71	<0.001
Alcohol and drugs	0.94	<0.001
Older adults (50+)	0.21	0.25
Lifestyle factors	0.02	0.92
FNPHC total	0.91	<0.001

## Data Availability

All data used in the manuscript are available upon request by contacting the lead author.

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
