# Peer review of "A Comparison of an Australian First Nations Primary Healthcare Data Specification with Potentially Preventable Hospitalisations"

_ijerph, 2024, doi:10.3390/ijerph21091192_

Round 1
Reviewer 1 Report
Comments and Suggestions for Authors
Dear editor, thank for for giving me an opportunity to review this manuscript. The manuscript is well-written and addresses an important issue in health care for Australian First Nations people. The research is very topical in Australian case and addresses the issue related to health inequality. The study's findings have significant implications for improving health care outcomes among First Nations people. I recommend to accept the manuscript. It would be nice if authors can address some following minor concerns.
- - Explicitly state why the current potentially preventable hospitalizations (PPH) specifications are insufficient for First Nations health care. Clearly articulating this gap will strengthen the rationale for the study.
- - Consider adding a sentence or two on how this research aligns with or builds upon previous studies in this area.
- - The discussion looks light in term of results. The study findings should be discussed more in term of availability evidence. Add more things on strength and policy implications. More references are required.
Many thanks,
Reviewer.
Author Response
Dear journal - we have uploaded a document that provides a response to all reviewers

Reviewer 2 Report
Comments and Suggestions for Authors
1.This paper is well-design and well-present, and provide useful information in healthcare.
2.How to create tables 1 -3, the author should ahve a detail description in the content related to teh data collection.
3.Discussion part seems a little weak to summarize the merit of this paper, it should have more discussion in this section.
Author Response
Dear journal - we have uploaded a document that provides a response to all reveiwers

Reviewer 3 Report
Comments and Suggestions for Authors
Referee Report for “A comparison of an Australian First Nations primary health care data specification with potentially preventable hospitalisations.”
· The authors should elaborate on the purpose of the paper, the contributions, and why this is important. Currently, these aspects are underdeveloped or missing altogether. For instance, is the main reason for the coding what is stated in lines 73-85 on page 2? If that’s one of the main motivations, the authors should explain the problems with using data from different resources and what synthesizing will accomplish. Note that the term synthesizing used in the paper is dubious. Does it mean that the authors will create alternative scales for ICD-10? Will they standardize existing scales so that they are comparable? Something different?
· Since the paper is specific to Australia, the authors should avoid using different titles for the same group (or include a footnote to state that they will use the terms interchangeably). For example, do the authors refer to non-Indigenous Australians as non-First Nations people or are these two different groups?
· The paper would benefit from providing a concrete example of how the authors got the results in Table 1. They can select a sample category and walk the reader through how they (re)classified any of these groups.
· Since the authors refer to the national average in Table 2, including it for each category and fiscal year would be beneficial.
· The Comparison with Potentially Preventable Hospitalisations section must include a more detailed discussion. How did the authors complete the comparison? Were there specific categories that were more different compared to others? How should we read Figure 1 and Figure 2?
· The same applies to the Principal and additional diagnoses section. The results are presented, but how do they tie back to the main topic of the paper? Also, the Discussion section needs a significant rewrite as it is just a repetition of what has been stated earlier. The Introduction had a good structure; the rest of the paper focuses on stating what tables and figures show but not on what the results mean.
· Both the Discussion and the Conclusion sections are incomplete.
Author Response

(The authors gave the same response as above.)

Round 2
Reviewer 2 Report
Comments and Suggestions for Authors
The author have revised the manuscript completely and appropriately, no more comments.